# AIRNET: A MACHINE LEARNING DATASET FOR AIR QUALITY FORECASTING

## ABSTRACT

In the past decade, many urban areas in China have suffered from serious air pollution problems, making air quality forecast a hot spot. Conventional approaches rely on numerical methods to estimate the pollutant concentration and require lots of computing power. To solve this problem, we applied the widely used deep learning methods. Deep learning requires large-scale datasets to train an effective model. In this paper, we introduced a new dataset, entitled as AirNet[1], containing the 0.25 degree resolution grid map of mainland China, with more than two years of continued air quality measurement and meteorological data. We published this dataset as an open resource for machine learning researches and set up a baseline of a 5-day air pollution forecast. The results of experiments demonstrated that this dataset could facilitate the development of new algorithms on the air quality forecast.

## 1 INTRODUCTION

In recent years, along with economic development, air pollution in developing countries such as China and India has become a severe problem threatening the public health(Pun et al. (2014), Lv et al. (2016)). One of the most abundant air pollutants PM2.5 (fine particles with a diameter 2.5 micrometers($\mu m$) or less) could penetrate the deepest part of the lungs such as the bronchioles or alveoli and result in asthma, lung cancer, respiratory diseases, cardiovascular disease etc.(contributors (2017)).

Air quality forecasting techniques are being rapidly upgraded as the demand for measuring pollution increases. Models like HYSPLIT-4 and KF utilize atmospheric dynamic processes and attempt to figure out the accumulation and dissipation mechanisms of air pollutants (Lv et al. (2015); Djalalova et al. (2015)). Hidden Markov model are also used to predict pollutant concentrations (Sun et al. (2013); Yetilmezsoy & Abdul-Wahab (2012)). However, due to the complexity of air transport dynamics, conventional forecasting models intrinsically demand a great amount of computing resources. In addition, conventional model's accuracy depends on the model structure itself and cannot improve regardless of the amount of training data.

On the other hand, the deep learning approach (LeCun et al.) has achieved exceptional results in unstructured information processing, such as computer vision, speech recognition, and natural language processing (Hinton et al. (2012); Zhang & LeCun (2015); Krizhevsky et al. (2012)).In those tasks, deep learning method has outperformed conventional machine learning methods. Inspired by this, people are trying to use deep learning models such as Recurrent Neural Network(RNN) and Long Short-Term Memory (LSTM), to perform meteorological forecasting (Shangzan et al. (2017)). Currently, its application includes estimation of precipitation probability and air pollution, etc..

Notably, to apply deep learning techniques effectively, a large-scale dataset is required to train a model (Liu et al. (2017)). A good dataset could greatly incent the industry to develop the new models and offer a unified assessment standard, as in the case of ImageNet (Deng et al. (2009)) in the computer vision field. However, to our knowledge, such dataset is absent in the air quality forecast field so far. To facilitate research and data collection, we processed and published an air quality dataset, AirNet, so as to fill this gap. Furthermore, we conducted experiments to validate the capability of this dataset, and set up a baseline for air pollution prediction on the AirNet dataset.

---

[1]AirNet: http://airnet.caiyunapp.com

Table 1: GFS Field Description

| NUMBER | FIELD | DESCRIPTION |
| --- | --- | --- |
| 001 | tmp | Temperature [K] |
| 002 | rh | Relative Humidity [%] |
| 003 | ugrd | U-Component of Wind [m/s] |
| 004 | vgrd | V-Component of Wind [m/s] |
| 005 | prate | Precipitation Rate [kg/m2/s] |
| 006 | tcdc | Total Cloud Cover [%] |

## 1.1 RELATED WORK

Shi et al. (2015) proposed RNN and convolutional LSTM to forecast the precipitation in future two hours, which they formalized as a spatio-temporal issue. The air quality forecast is similar to weather forecast, but two factors make air quality forecast more difficult and distinct from estimating precipitation; 1) the time span of air quality forecast is longer than weather forecast, the former often forecasts in four or five days, sometimes even goes beyond ten days. 2) additional influential factors must be considered in air quality forecast, such as the dynamics of air pollutants and the interaction with meteorological conditions. Modeling with AirNet dataset, the difference will be explicated in Section 5 below.

Ong et al. (2016) applied RNN to predict PM2.5 with environmental sensor data, which improved the results accuracy. Kurt & Oktay (2010)s research on forecasting air pollution with neural networks demonstrated the methods superiority and feasibility . Liang et al. (2015) released a dataset containing the value of PM2.5 which is only measured in Beijing. After this, Liang et al. (2016) published a larger dataset to analyze the pollutant factor in five cities of China. All datasets used above are point-wise data, which do not allow us to model in a spatially explicit manner. In addition, as Table 3 shows, these datasets are too small to train a deep neural network. Thus it became essential and urgent to set up a larger scale training dataset to enhance the accuracy of the forecast results.

## 1.2 CONTRIBUTION AND OVERVIEW

In this paper, we delivered a dataset, AirNet, containing more than two years of 6 indices of air quality data from 1498 stations, which is at least 40 times larger than most previous datasets. We set up a baseline based the LSTM model with AirNet dataset. The results demonstrated the effectiveness of the deep learning model for the air quality forecast.

## 2 DATASET EXTRACTION

### 2.1 DATASET COLLECTION

At first, we collected the data from China National Environmental Monitoring Center (CNEMC) which runs the 1498 monitor stations spreading across the whole country. Every station monitors air quality in real-time and reports concentration of the different air pollutants every hour. Therefore we wrote a spider to fetch the data from the Data-Publish platform[2]. Secondly, we gathered the meteorological data from the Global Forecast System (GFS) which contains the 6 meteorological condition features as in Table 1.

### 2.2 ALIGNMENT DATA

For air quality data from CNEMC, in each city, there are several monitoring stations of the real-time air pollutant concentration per hour.

The GFS data format is a 3-dimensional matrix, released by the National Oceanic and Atmospheric Administration (NOAA) every six hours. Every release contains the meteorological condition fea-

---

[2]URL: http://106.37.208.233:20035

tures forecasting for 10 days in every 3 hours, and 16 days in every 12 hours. For each meteorological condition features like TEMP, there are 1038240 values at one time point globally. We converted this data to a matrix which equals the volume of (180 * 4 + 1) * (360 * 4), where 180 is the radial latitude, 360 is the radial longitude, and 4 is the invert of the 0.25 resolution. Subsequently, we also obtained the global forecasting models geospatial information.

These two kinds of data sources are distinct. GFS data is a 3 dimensional matrix while the air quality data is a two dimensional matrix at a time point, where one dim is station ID, and the other is air quality indices. Therefore we had to align these two datasets. For each GFS data time point, we selected required air quality data for all stations in China and interpolated them into a matrix covering the whole country.

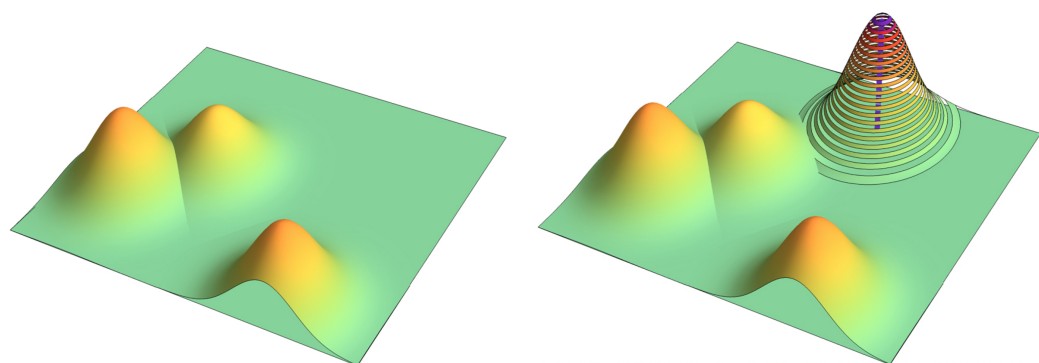

Figure 1: Use radial basis function to interpolate Air data from point-wise to a matrix.

Radial Basis Function is used to interpolate Air data to a matrix as shown in Figure 1. After that, we concatenated the gfs dataset with air quality dataset as in Figure 2 and thus produced a four-dimensional dataset (latitude, longitude, timesteps, features). Longitude ranges from 75 degrees to 132 degrees and the north latitude range of is from 18 degrees to 51 degrees. The grid resolution is 0.25 degree and the data was collected from April 1, 2015, to September 1, 2017. Every 3 hours there is one frame; on every frame we have 6 GFS features, and 6 types of the air quality indices. In total, we had a matrix with the dimension of (132, 228, 7072, 12). We took partial features of 2:00AM, January 23, 2017 as an example in Figure 2.

Some statistic features of the AirNet[3] are displayed in Table 3.

## 2.3   THE INTERPOLATED PRECISION

The error of dataset comes from two aspects, one is the error in the data collection process, and the other is the error of in the interpolation algorithm.

For air quality data from CNEMC, According HJ6, the measurement error is less than $10\mu g/m^3$. For meteorological data from NOAA, Quanzhi Ye validates the quality of Cloud data in Ye (2010), for example, the probability of below 30% forecast error is 63% for Paranal.

To validate the precision of interpolated value, we take off a proportion (10%) of real value in this dateset, then use these data to check the effect of interpolate algorithm. We choose the data from December 1, 2016 to December 31, 2016, and demonstrate the relative error and pearson correlation coefficientat different pollutant level, the result show as Table 2.

---

[3]BeijingPM25: https://archive.ics.uci.edu/ml/machine-learning-databases/00381/;
Five-Cities PM25: https://archive.ics.uci.edu/ml/machine-learning-databases/00394/

Table 2: The precision of interpolate algorithm. Every columns show interpolate precision at different pollution level.

|  | 0-50 | 51-100 | 101-200 | 201-300 | 301- | All |
|---|---|---|---|---|---|---|
| relative error | 0.4220 | 0.3041 | 0.3953 | 0.4691 | 0.4991 | 0.3816 |
| pearson correlation coefficient | 0.5697 | 0.4449 | 0.4222 | 0.2820 | 0.5062 | 0.7968 |

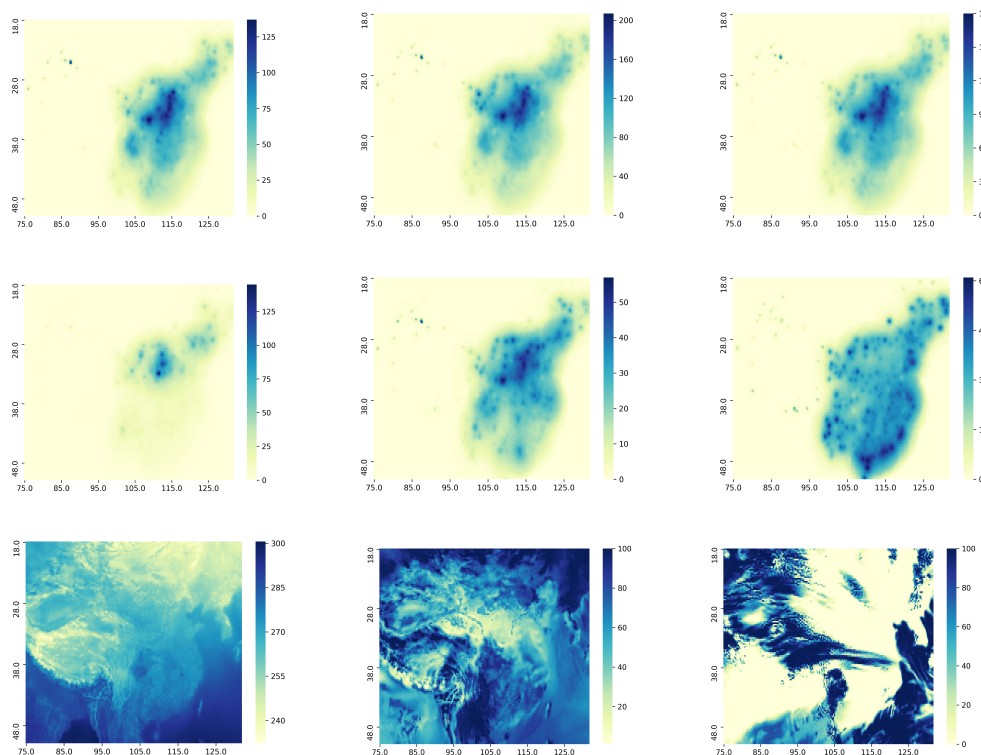

Figure 2: PM2.5, PM10, AQI, SO2, NO2, O3, tmp, rh, tcdc in 2:00AM January 23, 2017, mainland China, listed respectively from left to right and top to bottom, respectively.

## 3 MACHINE LEARNING TASK

### 3.1 TASKS DESCRIPTION

Air quality prediction is different from precipitation prediction because many factors will affect the concentration and distribution of the air pollutants. It is also essential to take the future meteorolog-

Table 3: Air pollutant dataset Characteristic

|  | Stations | Polutant | Time-span | Samples |
|---|---|---|---|---|
| AirNet | 1498 | PM2.5,PM10, NO2, CO, O3, SO2, AQI | 2015-4-1 2017-9-22 | 10593856 |
| BeijingPM25 | 2 | PM2.5 | 2010-1-1 2014-12-31 | 43824 |
| Five-Cities PM25 | 16 | PM2.5 | 2010-1-1 2015-12-31 | 262920 |

Table 4: Several concept about hit, miss and false alarm.

|  | prediction $>80$ | prediction $<= 80$ |
|---|---|---|
| fact $>80$ | hit | miss |
| fact $<= 80$ | false alarm | - |

ical conditions into consideration rather than just the history of pollutant concentration. Based on this observations, we formalized the pollutant problem as follows:

$$P\left(x_t\right) = P\left(x_t \mid x_{t-1}, x_{t-2} \ldots x_0; b_t, b_{t-1} \ldots b_0\right)$$
$$P\left(x_{t-1}, x_{t-2}, \ldots x_0\right) P\left(b_t, b_{t-1}, b_{t-2}, \ldots b_0\right)$$
(1)

Where the air pollutant concentration and meteorological condition at time step t are denoted as $x_t$ and $b_t$ respectively. We convert pollutant prediction into a sequential prediction problem. Since future meteorological conditions are as important as historical pollutant concentrations, the meteorological factors were taken as $b_0$ to $b_{t-1}$ plus the **predicted** future meteorological condition $b_t$. We could produce future meteorological predictions through numerical methods, and feed the predicted data into the model. As time t increments, the model described by formula 1 could be repeatly applied, so we could forecast air pollutant concentration as far into the future as possible.

## 3.2 METRICS

In the development of machine learning models, we use Mean Square Error as a loss function. Since the estimated pollutant value became more precise closer to the monitoring stations, we modified the MSE using a new calculation method, as the Point-wise Mean Square Error (PMSE), which only calculates loss in the nearby stations. The experiments results demonstrate that this improvement is evident and beneficial.

$$PMSE = \frac{1}{n}\sum_{i \in A}\left(\widehat{y}_i - y_i\right)^2 \qquad A := \{P \mid P \ located \ in \ a \ monitor \ station \ place\}$$
(2)

Additionally, we use the Probability of Detection (POD), the False Alarm Rate (FAR), and the Critical Success Index (CSI)Shi et al. (2015) as the metrics for the assessment. These metrics are intuitive for us to understand the performance of a system. We use PM2.5 value 80 ug/m3 as a threshold to discriminate polluted / non-polluted time steps (Table 4) and define the three indicators as:

$$POD = \frac{hits}{hits + misses + falsealarms}$$
(3)

$$FAR = \frac{falsealarms}{hits + falsealarms}$$
(4)

$$CSI = \frac{hits}{hits + misses}$$
(5)

## 4 BASELINE

For the pollutant prediction problem described in formula 1, we developed two baseline models. The first is a ReducedLSTM that captures pollutant accumulation and dissipation across sequential time steps at a given spatial location. The second is a convolutional model with learned location-specific kernels called WipeNet that also takes into account the transfer effect between different locations.

### 4.1 REDUCEDLSTM

We modify the original LSTM model (Hochreiter & Schmidhuber (1997)) to fit in the air pollutant change prediction in the field of air quality forecasting. The modification is demonstrated below.

The original LSTM model is:

$$
\begin{aligned}
f_t &= \sigma_g \left( W_f x_t + U_f h_{t-1} + b_f \right) \\
i_t &= \sigma_g \left( W_i x_t + U_i h_{t-1} + b_i \right) \\
o_t &= \sigma_g \left( W_o x_t + U_o h_{t-1} + b_o \right) \\
c_t &= f_t \circ c_{t-1} + i_t \circ \sigma_h \left( W_c x_t + U_c h_{t-1} + b_c \right) \\
h_t &= o_t \circ \sigma_h \left( c_t \right)
\end{aligned}
\tag{6}
$$

We modified this formula as follow:

$$
\begin{aligned}
f_t &= \sigma_g \left( W_f x_t + U_f h_{t-1} + b_f \right) \\
i_t &= W_i x_t + U_i h_{t-1} + b_i \\
c_t &= f_t \circ Relu \left( c_{t-1} + i_t \right) \\
h_t &= c_t - mean_t
\end{aligned}
\tag{7}
$$

The input and output gate were removed, as suggested in Jozefowicz et al. (2015). In the above formulas, $x_t$ represents the meteorological conditions, i.e., temperature, humidity, etc.. We regard $i_t$, as the accumulation of air pollutants induced by adverse weather conditions like higher temperature and no wind speed, and add it to $c_{t-1}$ to simulate the air pollutant accumulation. We use the forget gate to simulate the pollutant dissipation effect caused by factors like wind, rainfall, and high humidity.

At time step t, we stored the air pollutant concentration in $c_t$ and the subtract the moving average value which was calculated in advance to render $h_t$, $h_t$ and $c_t$ are then fed to the next time step.

### 4.2 WIPENET

In the above LSTM and ReducedLSTM method, we predict the future air pollutant concentration without considering meteorological conditions from nearby places. In the real physical environment, however, meteorological conditions, especially the flow of wind field, would impact pollutant concentration by transporting pollutant from one place to another.

In the WipeNet model, we consider the transport of pollutants as a redistribution process over spatial locations and use a locally connected convolutional operation with location-specific weights to model this process, as shown in Figure 3.

Sepcifically, after apply ReducedLSTM to an earlier geographical pollutant map as formula 7, we obtain $\widehat{c}$ as pollutant change on each geographic point at time step $t$, for clearly, we denote $\widehat{C} \in R^{D_h \times D_w}$ as the whole map value not $\widehat{c}_t$ at every point at this time step. Here, we define $D_h$ as the height of pollutant map and $D_w$ as the weight.

We feed wind field $Wind_t$ into a convolutional layer, to get a redistribution kernel component $KC_t$, consider two wind attribute like speed and direction, $Wind_t \in R^{D_h \times D_w \times 2}$ and $KC_t \in R^{D_h \times D_w \times k^2}$, here, $k$ is height and width of location-specific redistribution kernel.

Then we apply softmax to the channel dimension of $KC_t$ and reshape $KC_t$ into $R^{D_h \times D_w \times k \times k}$, thus obtaining a location-specific redistribution kernel.

Finally, we apply these redistribution kernels on $\widehat{C}$ to simulate atmosphere transportation effect. The update equations of WipeNet are given in Formula 8.

$$
\begin{aligned}
F_t &= \sigma_{hg}\left(W_f * X_t + U_f * H_{t-1} + B_f\right) \\
I_t &= Wi * X_t + U_i * H_{t-1} + B_i \\
\widehat{C}_t &= F_t \circ Relu\left(C_{t-1} + I_t\right) \\
RD_t &= reshape(softmax(W_k * Wind_t)) \\
C_t &= RD_t * \widehat{C}_t \\
H_t &= C_t - Moving\_avg_t
\end{aligned}
\tag{8}
$$

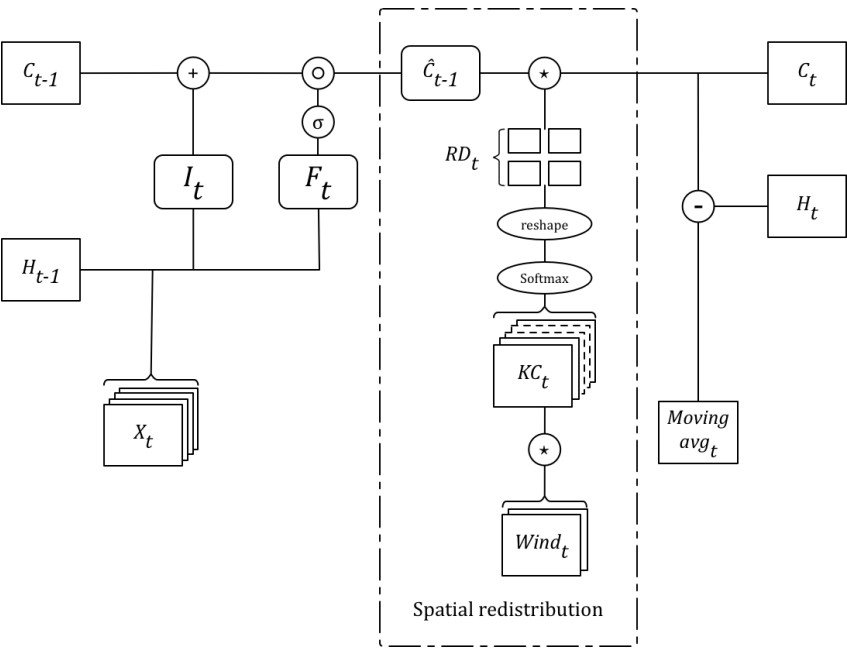

Figure 3: The model structure of WipeNet

## 5   EXPERIMENTS

In the previous study, Liang et al. (2015) demonstrated that pollutant concentration is significantly affected by meteorological conditions, hence we feed the meteorological condition at every time steps as the control information. Preliminarily, we found the LSTM outperformed the ReducedL-STM and Gated Recurrent Unit (GRU). We chose PM2.5 as the forecasting object of the air pollutant, and selected October 5, 2016, to January 3, 2017 data, as the training dataset, and January 3, 2017, to February 2, 2017,data as the validation dataset. As there is one data point every three hours, and we predict air pollutant concentration for 5 days, we sliced each dataset into 40 time steps length segment, allowing overlaying segments. In total, 680(90 * 8 - 40) matrix samples in the training dataset, and 200(30 * 8 - 40) matrix samples in the validation dataset were obtained, respectively. Because the methods of LSTM and ReducedLSTM did not use the spatial relationship, we shuffle our data on the locational dimension. For better precision, we chose the data points logged at monitor station in the HuaBei area and got 91120(for every matrix we got 134 stations) samples for training and 26800 samples for validation. The test dataset was set from October 5, 2015, to January 3, 2016.

Table 5: Experiment results of different network structures. Titles with a star (LSTM* and Re-ducedLSTM*) mean prediction was produced with only previous air pollutant data.

|  | TEST | | |
| --- | --- | --- | --- |
|  | POD | FAR | CSI |
| GRU | 0.43 | 0.47 | 0.31 |
| LSTM | 0.75 | 0.43 | 0.48 |
| ReducedLSTM | 0.63 | 0.43 | 0.44 |
| LSTM* | 0.38 | 0.53 | 0.23 |
| ReducedLSTM* | 0.49 | 0.54 | 0.27 |
| WipeNet | 0.76 | 0.30 | 0.56 |

We implemented all code through Keras (Chollet et al. (2015)), and chose TheanoBergstra et al. (2011) as backend. For simplification, we trained and predicted using subtracted PM2.5 data between two consecutive time steps. We preprocessed the meteorological condition information through two multiple perceptron (MLP), as each dimension of MLP is 20. We set the batch size to 32 and the dropout rate to 0.2, we ran 30 epoch with the patience option of 10. After every 10 experiments, the average value was calculated and the results are displayed in Table 5.

The results demonstrated that, without taking the meteorological condition into consideration, the accuracy of the prediction results dramatically deteriorated. Furthermore, ReducedLSTM is improved than LSTM in certain case, we assumed this is because our equation considered air pollutant dynamics. Furthermore, ReducedLSTM is better than LSTM in certain case, because we use more prior knowledge (eg. air pollutant dynamics) to design model than LSTM while keeping LSTM's advantage.

Finally, we implemented the WipeNet, with kernel size of (5, 5), and initialize weight with the Glorot Uniform Distribution, the dataset and the other settings were left unchanged.

We could see that the WipeNet achieved best accuracy on all of our three criterion. We attributed the progress to the integration of more information to the model, taking the transportation factor between different areas into consideration.

# 6 CONCLUSION

In this paper, we publicized a new dataset, AirNet, for researchers who want to use deep learning method to analyze air quality. Compared to previous studies in the field, it contains 6 indices of air quality from 1498 monitoring stations, which is at least 40 times larger than most previous datasets. In addition, we set up a baseline method WipeNet, for 5-day air quality prediction using AirNet dataset, and received a CSI score of 0.56, which achieved a 16% point improvement compared to classic LSTM methods.

## 6.1 FUTURE WORK

In the future, we plan to add more data types into AirNet, for example, we only used ground meteorological data in this paper, but data from multiple heights can reveal the change of inversion temperature layer which is a crucial factor for air quality forecast.

We wish AirNet would not only be applied to air quality forecast but also be utilized to reveal the critical factors in the causality of air pollution. For example, if we combine land-cover data with air pollution change, we may find some interactions between them. Perhaps we could even find new methods to reduce air pollution and give our children a brighter future.

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
