# OpenReview forum: "AirNet: a machine learning dataset for air quality forecasting"
_ICLR.cc/2018/Conference — Reject_

### Official Review · AnonReviewer3 · 2017-11-26
**It looks like the dataset is useful but the model development and experimental sections are weak.**

**Rating:** 5
**Confidence:** 4

**Review:**

The paper is about open sourcing AirNet, a database that has interpolated air quality metrics in a spatial form along with matching meteorological data obtained elsewhere. In addition, the paper also develops a few baseline methods and evaluated using standard metrics such as detection rate, false alarms etc. The work is original and significant from an applications point of view. It looks like the dataset is useful but the model development and experimental sections are weak.

Strengths:
- open source data set for air quality monitoring that is significantly better than existing ones.
- baseline models using standard methods including RNN.

Weaknesses:
- The air quality data is measured at point locations (stations) which are interpolated to obtain spatial data. There is no evaluation on this step to make sure the interpolated data indeed reflects truth.
- Experiments doesn't seem to be carefully done using hyper-parameter tuning/ cross-validation. The model results may be misleading.
- Writing and formatting needs to be improved. Some examples - "quality of air quality", "people attempted to apply deep learning", "in the computer vision field .", "Some people also used the hidden Makov model", "radial of longitude", "in 2:00AM, January 23". The paper in general was not easy to follow at many places.
- Is Table 3 incomplete with one box unlabeled?
- Figure 3 is not clear. It is suggested to follow standard notations to represent the RNN structure (see Jurgen Schmidhuber's paper)
- "DEV" in table 4 is not explained. Is this a development set? If so, what does it mean?
- It is said that "reduced LSTM is improved than LSTM". But the test results in Table 4 shows that LSTM is better.

---

> ### Author Response · Authors · 2017-12-31
> **Response to Reviewer 3**
>
> Thanks for your very constructive feedback. We have uploaded a revision to incorporate your suggestions. We will try to answer your questions and concerns one by one below.
>
> - We added a discussion on the accuracy of interpolation in section 2.3. Using data from 90% of monitoring stations, the predicted data were interpolated on a geographic coordinate grid of 0.25 degree across China. The correlation between the interpolated data and the remaining 10% of monitoring stations is 0.79. Researchers at Harvard University used satellite measurements of Aerosol Optical Depth (AOD), ground topography and so on to estimate pm2.5 in the area lacking of monitoring stations (Di (2017)). They obtained a coefficient of determination(r-squared) of 0.83. The accuracy of our interpolation is not much different from the results of using more data. So we think that the adjusted interpolation method for pm2.5 is still good enough for pm2.5 predictions. Thanks for your reminder, we will take better approaches to estimating pm2.5 values in the area lacking of monitoring stations in our future work.
> -------
> Di, Qian, et al. "Air pollution and mortality in the Medicare population." New England Journal of Medicine 376.26 (2017): 2513-2522.
>
>
> - Experiments doesn't seem to be carefully done using hyper-parameter tuning/ cross-validation. The model results may be misleading.
> Re: We used GRU (Gated Recurrent Unit) , LSTM, and reducedLSTM to test our dataset. We carefully tune parameters of those three models can get the best result we could. The result is listed in table 5.
>
> - Writing and formatting needs to be improved. Some examples - "quality of air quality", "people attempted to apply deep learning", "in the computer vision field .", "Some people also used the hidden Makov model", "radial of longitude", "in 2:00AM, January 23". The paper in general was not easy to follow at many places.
> Re: We fixed typos and grammar errors, and rewrotte the unclear sentences.
>
> - Is Table 3 incomplete with one box unlabeled?
> Re: Table 3 (now as Table 4) was completed, because we just used three kinds of counts to calculate POD, FAR and CSI. We added a short bar to indicate this box is not related.
>
> - Figure 3 is not clear. It is suggested to follow standard notations to represent the RNN structure (see Jurgen Schmidhuber's paper)
> Re:  Figure 3 is the model architecture of WipeNet, we redrew it and added reference in section 4.2
>
> - "DEV" in Table 4 is not explained. Is this a development set? If so, what does it mean?
> Re: DEV in Table4 (now as Table 5) means developing dataset. It’s useless to show the result and conclusion, so we removed it and just keep result in test set. Thanks you for reminder.
>
> - It is said that "reduced LSTM is improved than LSTM". But the test results in Table 4 shows that LSTM is better.
> Re: As you noticed, LSTM is better when adding meteorological data than ReducedLSTM as table 4 (now in table 5) shows. The sentence was wrong. We have fixed this bug.
> Furthermore, without meteorological data, ReducedLSTM outperforms LSTM. Consider its fewer parameters compared with LSTM, we think this is due to we use more prior knowledge to design the model.
>
> We hope that these replies and the revision resolve your questions. Any additional questions and suggestions are welcome and we will try our best to make things as clear as possible.

---

### Official Review · AnonReviewer2 · 2017-11-26
**This work offers a new air quality dataset. The authors describe the collection of data sources, and the processing of them. They then defined the machine learning task of predicting future air quality. Two baseline models are offered – ReducedLSTM and WIPENET. Experiments are done to demonstrate that this dataset is suitable to train deep models that predicts air quality.**

**Rating:** 4
**Confidence:** 4

**Review:**

The major contribution lies in the producing of the data. There are several concerns.
1. Since the major contribution lies in the production of the data, it is required for the authors to justify the quality of data. How accurate they are? What are the error bounds in terms of devices of measurement? What is the measurement precision? There is no such discussion for the data source in this submission, and thus it would be really hard for the reviewer to judge the validity of the dataset. The authors claim this is the largest dataset of such purpose, but they didn't demonstrate that the smaller datasets offered previously is indeed less competitive.

2. Using interpolation to align data is questionable. There are obviously many better ways to do so.

3. I would suggest the authors should use the two baseline models on other air-quality datasets for comparison. It can then convince the readers this dataset is indeed a better choice for the designed task.

4. This paper is not very well written. The English has certain room for improvement, and some details are missing. For instance, in Table1, Table2 and Table3, there are no captions . It is also unclear what's the purpose of Figure3 for?

---

> ### Author Response · Authors · 2017-12-31
> **Response to Reviewer 2**
>
> Thanks for your very constructive feedback. We have uploaded a revision to incorporate your suggestions. We will try to answer your questions and concerns one by one below.
>
> >1. Since the major contribution lies in the production of the data, it is required for the authors to justify the quality of data. How accurate they are? What are the error bounds in terms of devices of measurement? What is the measurement precision? There is no such discussion for the data source in this submission, and thus it would be really hard for the reviewer to judge the validity of the dataset. The authors claim this is the largest dataset of such purpose, but they didn't demonstrate that the smaller datasets offered previously is indeed less competitive.
>
> Re: Our data sources include CNEMC and NOAA. CNEMC uses the measured weight method to determine the air quality according to the standard HJB-2011, while NOAA uses many different method to measure different meteorological data. We added these information into section 2.3 as below:
> For air quality data from CNEMC, according to the decription of HJ6, the measurement error is less than 10µg/m 3 . For meteorological data from NOAA, Quanzhi Ye validates the quality of Cloud data in Ye (2010). For example, the probability of below 30% forecast error is 63% for Paranal..
> -------
> Ye, Q.-Z. 2011, Forecasting Cloud Cover and Atmospheric Seeing for Astronomical Observing: Application and Evaluation of the Global Forecast System, Publications of the Astronomical Society of the Pacific, 123, 113-124
>
> >2. Using interpolation to align data is questionable. There are obviously many better ways to do so.
> Re:   We added a discussion on the accuracy of interpolation in section 2.3. Using data from 90% of monitoring stations, the predicted data were interpolated on a geographic coordinate grid of 0.25 degree across China. The correlation between the interpolated data and the remaining 10% of monitoring stations is 0.79. Researchers at Harvard University used satellite measurements of Aerosol Optical Depth (AOD), ground topography and so on to estimate pm2.5 in the area lacking of monitoring stations (Di (2017)). They obtained a coefficient of determination(r-squared) of 0.83. The accuracy of our interpolation is not much different from the results of using more data. So we think that the adjusted interpolation method for pm2.5 is still good enough for pm2.5 predictions. Thanks for your reminder, we will take better approaches to estimating pm2.5 values in the area lacking of monitoring stations in our future work.
> -------
> Di, Qian, et al. "Air pollution and mortality in the Medicare population." New England Journal of Medicine 376.26 (2017): 2513-2522.
>
> >3. I would suggest the authors should use the two baseline models on other air-quality datasets for comparison. It can then convince the readers this dataset is indeed a better choice for the designed task.
> Re : Thanks for your advice. We do not have enough time to do those tests at the current stage of publicizing the datasets and the preliminary studies, and we will continue to do the comparison after the primary recognition from peers.
>
> >4. This paper is not very well written. The English has certain room for improvement, and some details are missing. For instance, in Table1, Table2 and Table3, there are no captions . It is also unclear what's the purpose of Figure3 for?
> Re: We rewrote this paper for better simplicity and clarity. The missing captions you mentioned were added as:
> Table1: GFS Field Description
> Table2: The precision of interpolate algorithm. Every columns show interpolate precision at different pollution level
> Table3: Air pollutant dataset Characteristic
> Table4: Several concept about hit, miss and false alarm.
>
> figure 3 is the model architecture of WipeNet. We redrew it and added reference in section 4.2
>
> We hope that these replies and the revision resolve your questions. Any additional questions and suggestions are welcome and we will try our best to make things as clear as possible.

---

### Official Review · AnonReviewer1 · 2017-11-27
**Main contribution of the paper is building a spatio-temporal data set on air pollution indicators from open source data and a baseline machine learning prediction model. The paper needs rewriting of certain of its paragraphs (bad phrasings and missing details).**

**Rating:** 4
**Confidence:** 4

**Review:**

This paper's main contribution is in the building of a spatio-temporal data set on air pollution indicators as the title states.
The data set is built from open source data to comprise pollutants measured at a number of stations and meteorological data. Then, an air pollutant predictor is built as a baseline machine learning model with a reducedLSTM model.
Most of the first part's work is in the extraction of the public data from the above mentioned sources, aligning of the two source data and sampling considerations.
The paper lacks detailed explanation of the problem it is actually addressing by omitting the current systems' performance: simply stating: 1.1/page 2 "Thus it became essential and urgent to set up a larger scale training dataset to enhance the accuracy of the forecast results."
It also lacks definition of certain application domain area terms and acronyms (PM2:5).
Certain paragraphs need rewriting:
     - 2.2/Page 3: "Latitude ranges from 75 degrees to 132 degrees and the north latitude range of is from 18 degrees to 51 degrees".
     - 3.1/Page 4: "We converted the problem of the pollutant prediction as time sequential prediction problems, as in the case of giving the past pollutant concentration x0 to xt􀀀1.".
Also, Table 1: GFS Field Description contains 6 features  not 7 as stated in 2.1

For air pollutant prediction a baseline machine learning model is built with a reducedLSTM model.
Results seem promising but lack serious comparison with currently obtained results by other approaches as mentioned above. The statement in 5./Page 7:"Furthermore, reduced LSTM is improved than LSTM, we assumed this is because our equation considered air pollutant dynamics, thus we gave more information to model than LSTM while keeping LSTMs advantage." attributes the enhanced results to extra data (quantity) fed to the model rather than the fact (quality) as stated in the paper that the meteorological conditions (dispersion etc.) influence the air pollutant presence/ concentrations in nearby stations.
A rewriting and clarification of certain paragraphs is therefore recommended.

---

> ### Author Response · Authors · 2017-12-30
> **Response to Reviewer 1**
>
> Thanks for your very constructive feedback. We have uploaded a revision to incorporate your suggestions. We will try to answer your questions and concerns one by one below.
>
> >The paper lacks detailed explanation of the problem it is actually addressing by omitting the current systems' performance: simply stating: 1.1/page 2 "Thus it became essential and urgent to set up a larger scale training dataset to enhance the accuracy of the forecast results."
>
> Re:  Chinese Ministry of Environmental Protection (CMEP) is currently providing air quality forecasts for the next two days, but CMEP does not publish the statistical data of their accuracy of forecast, so we didn't put it into our paper. In our practical experience, the accuracy is not good. One of the air quality pollution warning was released on 27 October 2017 in Beijing, leading many people to cancel the outdoor activities, but the pollution did not come as predicted. Our experience makes us feel that the analytical work of air quality is very important, particularly considering that machine learning methods have developed so quickly. If there are public datasets, there is a good chance that researchers can build more accurate forecasting models for longer periods and will greatly improve the lives of the public.
>
> >It also lacks definition of certain application domain area terms and acronyms (PM2:5).
> Re: We have added more background information into introduction section. The PM2.5's definition and its danger have been added into first paragraph: One of the most abundant air pollutants PM2.5 (ﬁne particles with a diameter 2.5 micrometers(µm) or less) could penetrate the deepest part of the lungs such as the bronchioles or alveoli and result in asthma, lung cancer, respiratory diseases, cardiovascular disease etc.
>
> >Certain paragraphs need rewriting:
> >    - 2.2/Page 3: "Latitude ranges from 75 degrees to 132 degrees and the north latitude range of is from 18 degrees >to 51 degrees".
> Re: Changed to "Longitude ranges from 75 degrees to 132 degrees and the north latitude range of is from 18 degrees to 51 degrees."
> >     - 3.1/Page 4: "We converted the problem of the pollutant prediction as time sequential prediction problems, as in >the case of giving the past pollutant concentration x0 to xt􀀀1.".
> Re:  We rewrite 3.1 to make it clearer.
> >Also, Table 1: GFS Field Description contains 6 features  not 7 as stated in 2.1
> Re: It's 6 features, we've corrected it in 2.1
>
> >Results seem promising but lack serious comparison with currently obtained results by other approaches as >mentioned above.
> Re: We used GRU (Gated Recurrent Unit) , LSTM, and reducedLSTM to test our dataset. We carefully tuned parameters of those three models to get the best possible results. The result is listed in table 5.
>
> >The statement in 5./Page 7:"Furthermore, reduced LSTM is improved than LSTM, we assumed >this is because our equation considered air pollutant dynamics, thus we gave more information to model than LSTM >while keeping LSTMs advantage." attributes the enhanced results to extra data (quantity) fed to the model rather >than the fact (quality) as stated in the paper that the meteorological conditions (dispersion etc.) influence the air >pollutant presence/ concentrations in nearby stations.
> Re: The original sentence was ambiguous. It was aiming to emphasize that we used more prior knowledge to improve robustness of the model. We rewrote this paragraph to disambiguate as follow: ReducedLSTM is better than LSTM in certain case, because we use more prior knowledge (eg. air pollutant dynamics) to design the model than LSTM while keeping LSTM's advantage.
>
> We hope that these replies and the revision resolve your questions. Any additional questions and suggestions are welcome and we will try our best to make things as clear as possible.

---

### Author Response · Authors · 2018-01-04
**Revision 2017-12-31: Summary of Changes**

We have revised our paper based on the reviewers' comments. All of our changes are summarized below:
1) we have corrected grammatical and spelling errors.
2) we have added defination of PM2.5.
3) we have added section 2.3 to show the precision of our interpolate algorithm.
4) we have rewritten section 4.2 to make the architecture of WipeNet clearer.
5) we have fixed some logic errors like the inconsistency betweent expression and table about the comparison of LSTM and ReducedLSTM.

---

### Decision · Program_Chairs · 2018-01-29
**ICLR 2018 Conference Acceptance Decision**

**Decision:**

Reject

**Comment:**

This is an interesting application area, but the quality of the presentation and experimental work here is not sufficient for acceptance. The numerical ratings from reviewers are just not high enough to warrant acceptance.